# Antimutagenic, Cytoprotective and Antioxidant Properties of *Ficus deltoidea* Aqueous Extract In Vitro

**DOI:** 10.3390/molecules26113287

**Published:** 2021-05-29

**Authors:** Theng Choon Ooi, Farah Wahida Ibrahim, Shakirah Ahmad, Kok Meng Chan, Lek Mun Leong, Nihayah Mohammad, Ee Ling Siew, Nor Fadilah Rajab

**Affiliations:** 1Center for Healthy Aging and Wellness, Faculty of Health Sciences, Universiti Kebangsaan Malaysia, Jalan Raja Muda Abdul Aziz, Kuala Lumpur 50300, Malaysia; ooithengchoon@ukm.edu.my (T.C.O.); kierah.ahmad@gmail.com (S.A.); 2Center for Toxicology and Health Risk Studies, Faculty of Health Sciences, Universiti Kebangsaan Malaysia, Jalan Raja Muda Abdul Aziz, Kuala Lumpur 50300, Malaysia; farahwahida@ukm.edu.my (F.W.I.); chan@ukm.edu.my (K.M.C.); nihayah@ukm.edu.my (N.M.); 3Prima Nexus Sdn Bhd, Suite 8-1 Level 8, Menara CIMB, Jalan Stesen Sentral 2, Kuala Lumpur 50470, Malaysia; leonglmn@gmail.com; 4ASASI Pintar Program, Pusat GENIUS@Pintar Negara, Universiti Kebangsaan Malaysia, Bangi 43600, Malaysia; sieweeling@ukm.edu.my

**Keywords:** Ames, antimutagenicity, antioxidant, cytoprotection, *Ficus deltoidea*

## Abstract

*Ficus deltoidea* var. *deltoidea* is used as traditional medicine for diabetes, inflammation, and nociception. However, the antimutagenic potential and cytoprotective effects of this plant remain unknown. In this study, the mutagenic and antimutagenic activities of *F. deltoidea aqueous extract* (FDD) on both *Salmonella typhimurium* TA 98 and TA 100 strains were assessed using *Salmonella* mutagenicity assay (Ames test). Then, the cytoprotective potential of FDD on menadione-induced oxidative stress was determined in a V79 mouse lung fibroblast cell line. The ferric-reducing antioxidant power (FRAP) assay was conducted to evaluate FDD antioxidant capacity. Results showed that FDD (up to 50 mg/mL) did not exhibit a mutagenic effect on either TA 98 or TA 100 strains. Notably, FDD decreased the revertant colony count induced by 2-aminoanthracene in both strains in the presence of metabolic activation (*p* < 0.05). Additionally, pretreatment of FDD (50 and 100 µg/mL) demonstrated remarkable protection against menadione-induced oxidative stress in V79 cells significantly by decreasing superoxide anion level (*p* < 0.05). FDD at all concentrations tested (12.5–100 µg/mL) exhibited antioxidant power, suggesting the cytoprotective effect of FDD could be partly attributed to its antioxidant properties. This report highlights that *F. deltoidea* may provide a chemopreventive effect on mutagenic and oxidative stress inducers.

## 1. Introduction

Cancer is a major health burden to society in both developed and developing countries [1]. Cancer was the second major cause of death in Malaysia in 2014 [2]. The causes of some cancers, for instance colorectal cancer, can be hereditary in nature, but most cases are inexplicable [3]. Cancer is often related to mutations caused by mutagens.

Mutagens are chemical or physical agents that can alter the genetic content of an organism. Mutations in germ line cells can be passed to future generations, whereas mutations in somatic cells can lead to various pathological conditions, such as cancer [4]. Mutation can occur as point mutation, large deletion, rearrangement of DNA, chromosome break, or gain or loss of whole chromosomes [5]. Of these mutation types, point mutation is the most common DNA sequence alteration [4,6].

To address the deleterious effect of mutagenicity, scholars aim to identify antimutagens, which are a group of chemicals that can decrease mutation rates [4,7]. Antimutagens can be categorized into the following based on their mechanism of action: antimutagens with antioxidant activity, antimutagens that inhibit the activation of mutagens, antimutagens that act as blocking agents, and antimutagens with multiple modes of action [4]. The search of antimutagens in dietary food and natural products revealed a number of promising antimutagens and provided insights into the benefits of some food over others [4,8]. Some herbs used as folk medicine were scientifically proven to possess antimutagenic properties [4,9,10,11,12]. Some medicinal herbs used in Southeast Asia with proven antimutagenic properties include *Clinacanthus nutans*, *Puerarie mirifica*, *Puerarie lobata*, and *Euphorbia hirta* [13,14,15].

Reactive oxygen species (ROS) are reactive molecules produced in living cells during normal cell metabolism or obtained from exogenous sources. These species include superoxide anion radicals (·O_2_^−^), hydrogen peroxide (H_2_O_2_), and hydroxyl radicals (·OH). The main endogenous sources of ROS in organisms are mitochondria, peroxisomes, and lysosomes. The exogenous sources of ROS include xenobiotics, tobacco smoke, organic solvents, and pesticides [16]. Oxidative stress is a state of imbalance caused by an excess amount of ROS or oxidant over that of antioxidant in cells. This type of stress is associated with various diseases, such as cancer, diabetes, neurodegeneration, and aging [16]. Therefore, maintaining redox balance is important for cellular homeostasis.

Cells must have a defense mechanism to effectively counteract oxidative stress. Cellular defense against ROS can be executed through an intracellular antioxidant mechanism and with the help of externally obtained antioxidants [17]. An antioxidant defense system in cells elicits protection against oxidative stress by stimulating the expression of phase I and phase II detoxification enzymes, such as glutathione, glutathione peroxidase, glutathione-S-transferase, superoxide dismutase, and catalase [18]. γ-l-glutamyl-l-cysteinyl-glycine, also known as glutathione (GSH), is one of the most important tripeptides in regulating redox status because of its ability to exist as different redox species. As a redox signaling modulator, GSH effectively scavenges free radicals either directly or indirectly through enzymatic reactions [19].

*Ficus deltoidea* is a traditional medicinal plant widely distributed in Malaysia, Thailand, and Indonesia. This plant is commonly known as *Mas Cotek*, *Telinga Beruk*, or *Serapat Angin* in Malaysia; as *Sempit-Sempit* and *Agoluran* in Sabah, Sarawak, and Kalimantan Islands; and as *Tabat Barito* in Indonesia. This plant has a wide ethnopharmacology application among Malays. The decoction of *F. deltoidea* leaves is used to treat diabetes, menstrual cycle disorder, and leucorrhoea, and serves as a tonic to contract uterus and vagina muscles after birth [20,21,22].

In this study, we investigated the mutagenic, antimutagenic, and cytoprotective potentials of *F. deltoidea* var. *deltoidea* aqueous extract (FDD). *Salmonella*/mammalian microsome test (Ames test) was used to determine the mutagenic and antimutagenic properties of FDD. The cytoprotective potential of FDD was also assessed in V79 Chinese hamster lung fibroblasts treated with menadione, a redox cycling agent. This study aims to explore the benefits of FDD and improve our scientific knowledge on the medicinal properties of this popular natural product in Southeast Asia.

## 2. Results

### 2.1. Mutagenicity of FDD

FDD was subjected to *Salmonella* mutagenicity testing (Ames test) using two *Salmonella typhimurium* strains, namely TA 98 and TA 100. Testing was carried out in two conditions, namely, with and without metabolic activation. Revertant colony count did not significantly increase relative to that of the negative control under both conditions; hence, the result showed that FDD was not mutagenic at all tested concentrations (3.125–50 mg*/*mL) (Table 1). Considering the absence of mutagenic effect in the Ames test without metabolic activation, we only applied metabolic activation to the lowest (3.125 mg/mL), middle (12.5 mg/mL), and highest (50 mg/mL) concentrations.

### 2.2. Antimutagenicity of FDD

The antimutagenic effect of FDD was determined in this study (Table 2). In the absence of metabolic activation, 50 mg/mL FDD showed weak inhibition in TA 98 strain, whereas 12.5 and 50 mg/mL FDD showed no antimutagenic effect on TA 100 strain. Interestingly, 50 mg/mL FDD exhibited a strong antimutagenic effect on both strains in the presence of metabolic activation; revertant colony count significantly decreased compared with that of the positive control (*p* < 0.05). Our data showed that FDD was able to inhibit mutation induced by 2-aminoanthracene in the presence of metabolic activation.

### 2.3. Cytotoxicity of FDD

In this study, cytotoxicity of FDD in V79 cells was determined before in vitro testing was conducted. Treatments of up to 500 µg/mL FDD for 24 h did not decrease cell viability. Hence, 25–100 µg/mL FDD was selected as the safe concentration range for further testing (Figure 1).

### 2.4. Cytoprotective Effect of FDD

In this study, the protective effects of FDD against menadione-induced oxidative damage were evaluated. Pre-treatment of FDD (50 and 100 µg/mL) for 24 h significantly decreased intracellular superoxide anion concentration produced by menadione (*p* < 0.05). Hence, FDD pre-treatment protected against menadione-induced superoxide anion accumulation in V79-4 cells (Figure 2). We examined whether FDD pre-treatment elicited protection against cell death as a later event in oxidative stress. Figure 3 shows that only 100 µg/mL FDD conferred partial protection against cell death induced by menadione. FDD (100 µg/mL pre-treatment group) increased cell viability from 61.7 ± 3.9% in the menadione alone group to 75.8 ± 4.0%. However, the increase in cell viability was not statistically significant (*p* > 0.05).

### 2.5. FDD Modulation on GSH Level

In this study, we examined if FDD was able to increase GSH levels, an important component in intracellular antioxidant defense systems to confer cytoprotective effects. Figure 4 shows the GSH levels of different treatment groups of FDD. Our data demonstrated that FDD did not increase GSH levels in menadione-induced cells for all tested concentrations. Hence, the FDD protective mechanism was independent of GSH modulation.

### 2.6. Ferric-Reducing Antioxidant Power of FDD

In the FRAP assay, antioxidant in the sample is assumed as reducing agent and the reducing power of the antioxidant in a redox linked colorimetric reaction is measured. All ranges of FDD tested contained antioxidant power. The FRAP value of FDD was 9.64 ± 0.08 µg ascorbic acid equivalent/100 µg FDD or 150.67 ± 1.18 µM Fe^2+^, equivalent/100 µg FDD (Figure 5).

## 3. Discussion

In this study, *Salmonella* mutagenicity testing was employed to determine the potential mutagenic effect of FDD. This screening testing was recommended by the scientific community and government agencies for detecting chemically induced mutagenesis [5]. The chemicals tested are classified as mutagenic if they produce more than a twofold increase in the number of revertant colonies over that of the negative control or demonstrated a dose-related increase in revertant colonies. For all concentrations tested (3.125–50 mg/mL), none of the concentrations produced a twofold increase in revertant colony count or yielded a dose-related increase in colony count. Similar results were obtained when metabolic activation was introduced by adding S9 mix. Therefore, FDD was considered as non-mutagenic based on the provided classification [5]. Our data are in line with previous study on *F. deltoidea* methanolic extract, which showed no mutagenic activity [23]. However, this is just preliminary, and a further battery of tests is needed in terms of genetic toxicity data for public assurance of the safety of the plant used. Safety data of this popular plant is important because prepared decoction of *F. deltoidea* is commonly consumed in traditional medicine practice.

The antimutagenic activity of FDD was also screened against known mutagens by using *Salmonella* TA 98 and TA 100 strains. In the absence of S9 metabolic activation, both concentrations of FDD (12.5 and 50 mg/mL) did not show any antimutagenic activity; however, 50 mg/mL FDD exhibited weak antimutagenic activity on the TA 98 strain. FDD inhibition rate slightly increased with increasing extract concentration. In the presence of S9 metabolic activation, FDD demonstrated a dose-dependent increase in antimutagenic activity. FDD (50 mg/mL) produced strong antimutagenic activity against 2-aminoanthracene in both TA 98 and TA 100 strains. Our results demonstrated that FDD performed poorly in inhibiting direct-acting mutagens in the assay without S9 metabolic activation, but could inhibit mutagenicity induced by S9 dependent (indirect-acting) mutagens in a dose-dependent manner. There are several possible explanations for the obtained result. Firstly, FDD may modulate liver-metabolizing enzymes, and thus prevent the transformation of 2-aminoanthracene to ultimate toxicant or promote the metabolism of 2-aminoanthracene to improve its excretion. Another possible explanation is that FDD may require modification by liver enzymes before producing metabolites with antimutagenic activity [14]. This study is the first to demonstrate the antimutagenic activity of *F. deltoidea* in the indirect-acting mutagen 2-aminoanthracene. The presented preclinical data provide novel evidence about the potential nutritional value of this popular plant relevant to consumers and improved scientific knowledge on the ethnopharmacology of *F. deltoidea*.

In this study, the protective effect of *F. deltoidea* against the potent ROS inducer menadione was investigated. Menadione is a vitamin K analog and is also known as vitamin K_3_ [24,25]. Menadione is a type of naphthoquinone that contains quinone moiety, which can generate ROS through redox cycling [24]. Menadione was marketed between 1939 and 1941 and used in clinical settings as a treatment for cholemic bleeding caused by biliary tract obstruction or fistula, hypopro-thrombinemia associated with other malabsorption syndromes, and hemorrhagic disease in newborns [25]. Menadione, as well as other chemicals that contain quinone moiety, has been studied in terms of its ROS-producing mechanisms via redox cycling [24]. In cells, the quinone moiety undergoes a reduction in the presence of flavoenzymes to form a semiquinone moiety. In the presence of oxygen molecules, the semiquinone moiety donates an electron to an oxygen molecule, thereby forming a superoxide radical anion (**·**O_2_^−^) and the original quinone. The regenerated quinone may undergo the same futile cycling to produce semiquinone. This process continues until the system becomes anaerobic [24,26]. During redox cycling, large amounts of superoxide radical anions are produced. Accumulation of ROS causes an imbalance in redox status and generates oxidative stress. Free radicals can oxidize cellular macromolecules, such as proteins, DNA, carbohydrates, and lipids, leading to cellular damage [24,27].

Our data demonstrated that 50 and 100 µg/mL FDD pre-treatment decreased superoxide anion levels generated by menadione. FRAP assay demonstrated that both concentrations possessed reducing properties, thereby indicating the antioxidant power of FDD. These data suggest that the ability of FDD to decrease superoxide anions generated by menadione could be partially attributed to the antioxidant properties of this plant. Aside from a direct antioxidant effect, the plant extract may modulate intracellular antioxidant responses, such as enzymatic and non-enzymatic antioxidant mechanisms, to confer protection against oxidative stress. In this study, we measured GSH content of V79 cells treated with menadione. Pre-treatment of FDD did not significantly affect GSH levels in cells. These data suggested that FDD confer protection against oxidative stress independent of GSH modulation. The possibility of FDD to confer protective effects via modulation of enzymatic pathways remains unclear and requires further investigation.

## 4. Materials and Methods

### 4.1. Sample Collection and Preparation

*F. deltoidea* var. *deltoidea* leaves were obtained from Juaseh Tengah, Negeri Sembilan, Malaysia. The plant samples were taxonomically identified and authenticated by the Herbarium, Faculty of Sciences and Technology, Universiti Kebangsaan Malaysia, with the voucher number UKMB 29781.

The *F. deltoidea* var. *deltoidea* leaves (250 g) were cut into small pieces and extracted with distilled water using a Soxhlet apparatus for 16 h. The extract was filtered and freeze dried to remove solvent residue. The extraction yielded 7.45 g of FDD, which accounted for a 2.98% yield. FDD was stored in an air-tight container and stored under chilled conditions (4 °C).

### 4.2. Salmonella Mutagenicity Assay

The *Salmonella* mutagenicity assay was performed using a previously described method with slight modifications [28]. *Salmonella typhimurium* TA 98 strain (frame shift mutation) and TA 100 strain (base pair substitution) were used in the assay. Both *Salmonella typhimurium* histidine auxotroph strains TA 98 (*his* 3052, *uvrB*, pKM101, *rfa*) and TA 100 (*his*G46, *uvrB*, pKM101, *rf*a) were obtained from the American Type Culture Collection (ATCC, Manassas, VA, USA). The mutant strains were confirmed for their genotypes of histidine/biotin dependence, *rfa* marker *uvrB* gene deletion mutation, and the presence of plasmid *pMK101*. This assay employed the plate incorporation method with and without exogenous metabolic systems. Test strains from frozen cultures were incubated in Oxoid nutrient broth No. 2 (Oxoid Ltd., Basingstoke, Hants, UK) overnight at 37 °C in a shaking water bath before use.

Briefly, FDD was reconstituted in sterilized normal saline to prepare a series of concentrations (3.125, 6.25, 12.5, 25, and 50 mg/mL). Different concentrations of FDD were added to 2 mL of top agar, supplemented with 0.5 mM l-histidine and 0.5 mM *d*-biotin. These mixtures were added with 100 µL of bacterial culture and poured onto a plate containing minimum agar. The plates were then incubated for 48 h, and his^+^ revertant colonies were counted after the incubation period. The influence of metabolic activation was tested by adding 500 µL of S9 mixture (Moltox, Boone, NC, USA). Sterilized normal saline was used as negative control to determine spontaneous reversion activity. For testing in the absence of metabolic activation, 2-nitrofluorene and sodium azide were used as positive control for TA 98 and TA 100 strains, respectively. For testing in the presence of metabolic activation, 2-aminoanthracene was used as positive control for both strains. The extract was classified as mutagenic if it produced more than a twofold increase in the number of revertant colonies compared with that of the negative control, or demonstrated a dose-related increase in revertant colonies.

### 4.3. Antimutagenicity Assay

This assay is based on the plate incorporation method, similar to *Salmonella* mutagenicity assay as described in Section 4.2. Briefly, 100 μL of FDD (12.5 and 50 mg/mL) was added to 2 mL of top agar supplemented with 0.5 mM l-histidine and 0.5 mM d-biotin. The mixture was then mixed with 100 µL of mutagen and 100 µL of bacterial culture (TA 98 and TA 100 strains) and poured onto minimum agar. For testing in the absence of metabolic activation, 2-nitrofluorene and sodium azide were used as mutagens for TA 98 and TA 100 strains, respectively. For testing in the presence of metabolic activation, 2-aminoanthracene was used as a mutagen for both strains. The plates were then incubated at 37 °C for 48 h, and his^+^ revertant colonies were counted after incubation. Percentage of inhibition was calculated based on the following Equation (1)
(1)$$\mathrm{Inhibition}\;\mathrm{rate}\;\left(\%\right)=100\;-\left[\left(\frac{T}{M}\right)\;\times \;100\%\right]$$
where *T* is the number of revertants per plate in the presence of both the mutagen and FDD, and *M* is the number of revertants per plate in the presence of the mutagen alone. The percentage of inhibition was classified as follows: higher than 60%, strong; 60% to 41%, moderate; 40% to 21%, weak; and < 20%, negligible [29].

### 4.4. Cell Culture

V79 Chinese hamster lung fibroblasts were purchased from ATCC (Rockville, MD USA). The cells were maintained under physiological conditions (37 °C, 5% CO_2_) in Dulbecco’s Modified Eagle’s Medium (Gibco, Grand Island, NY, USA) supplemented with 10% fetal bovine serum (PAA Laboratories, Morningside, Queensland, Australia) and 100 U/mL penicillin–streptomycin (PAA Laboratories, Morningside, Queensland, Australia). Prior to all testing in Subsections 2.8 to 2.11, the cells were seeded at a density of 1 × 10^4^/cm^2^ and incubated overnight to facilitate attachment. The cells were given treatment as described in each subsection.

### 4.5. MTT Assay

The viability of V79 cells was determined by using 3-4,5-dimethylthiazol-Z-yl-2,5-diphenyltetrazolium bromide (MTT) assay, as previously described [30]. Briefly, V79 cells were seeded in a 96-well plate and treated with 62.5–500 µg/mL FDD for 24 h to determine the effect of FDD on the viability of V79 cells. On the other hand, to determine the cytoprotective effect of FDD, V79 cells were seeded and treated with FDD (25–100 µg/mL) for 24 h prior to induction with 26 µM menadione for another 24 h. At the end of the treatment period, 20 µL of the MTT solution (5 mg/mL) was added to each well and incubated for 4 h before the medium was discarded. Subsequently, 200 µL of DMSO was added to dissolve the formazan crystals. The plates were shaken for 5 min to dissolve crystals formed, and absorbance was measured at 570 nm by using a microplate reader (Bio-Rad Laboratories, Hercules, CA, USA). Cell viability was calculated relative to the vehicle control group.

### 4.6. Superoxide Anion Assessment

Superoxide anion assessment was performed as described previously [31]. Briefly, V79-4 cells were seeded and treated with FDD (25–100 µg/mL) for 24 h. The cells were then treated with 26 µM menadione for 3 h to induce oxidative stress. After the treatment period, the cells were harvested through trypsinization, washed with chilled PBS, and suspended in pre-warmed serum-free medium at a concentration of 2 × 10^5^ cell/mL. The cells were then added to 1 μL of 10 mM hydroethidine (HE) and incubated in the dark (37 °C, 30 min). The cells were then centrifuged at 2500 rpm for 5 min to obtain cell pellets. The pellets were washed with chilled PBS, suspended in 500 µL of chilled PBS, and analyzed with a FASCantoII flow cytometer (BD Biosciences, San Jose, CA, USA).

### 4.7. Reduced Glutathione Quantification

The GSH level was measured as described previously [31]. Briefly, V79-4 cells were seeded and pre-treated with FDD (25–100 µg/mL) for 24 h. The cells were then treated with 26 µM menadione for 3 h to induce oxidative stress. The treated cells were collected by trypsinization and subjected to centrifugation (5 min, 2500 rpm) to collect cell pellets. The pellets were washed with PBS twice and added with 100 µL of cold lysis buffer (50 mM K_2_HPO_4_; 1 mM EDTA, pH 6.5; 0.1% *v*/*v* Triton X-100). The cells were incubated in ice for 15 min with gentle vortex every 5 min to facilitate lysis. The mixtures were centrifuged (10,000 rpm, 15 min, 4 °C), and supernatant was collected as protein lysate. Glutathione (GSH) standard curve was constructed using reduced GSH solution ranging from 1.25 mM to 0.125 µM. Briefly, 50 µL of the lysate or GSH standard solution was transferred into 96-well plates. Subsequently, 40 μL of the reaction buffer (0.1 M Na_2_HPO_4_, 1 mM EDTA, pH 8) and 10 µL of 4 mg/mL dithiobis (2-bitrobenzoic acid) (DTNB) were added into wells containing the sample and the standard. The plates were incubated for 15 min at 37 °C before the OD was read under 415 nm. Free thiol concentration was calculated based on the GSH standard and expressed as a nmol GSH/mg protein.

### 4.8. Ferric-reducing Antioxidant Power (FRAP) Assay

Ferric-reducing antioxidant power (FRAP) assay was performed using previously described methods with slight modification [32,33]. The assay was conducted in a microtiter plate format. FRAP reagent was prepared by mixing 300 mM acetate buffer (pH 3.6), 20 mM ferric (III) chloride, and 10 mM Tri(2-pyridyl)-s-triazine (TPTZ) in a ratio of 10:1:1 (*v*:*v*). FRAP reagent was prepared freshly and incubated at 37 °C for 15 min prior to use. Various concentrations of FDD (25–100 µg/mL), ascorbic acid (12.5–50 µg/mL), and ferric (II) sulfate (100–1000 µM) solution were prepared. Subsequently, 50 µL of each solution was loaded into a 96-well plate. Ascorbic acid and ferric (II) sulfate were used as standard in this assay. Each well was then added with 175 µL of FRAP reagent and incubated in the dark (room temperature, 5 min). Absorbance was read at 590 nm wavelength by using a microplate reader (Bio-Rad Laboratories, Hercules, CA, USA). The FRAP value of FDD was expressed as ascorbic acid equivalent (µg/mL).

### 4.9. Statistical Analysis

All experiments were conducted in triplicate, and the results were expressed as mean ± standard error mean for all experimental groups. Data were subjected to statistical comparison using one-way ANOVA (SPSS version 22, IBM Corp., Armonk, NY, USA). Values at *p* < 0.05 were considered significant.

## 5. Conclusions

FDD demonstrated no mutagenic activity on *Salmonella* TA 98 and TA 100 strains with and without metabolic activation. However, it demonstrated strong inhibition on 2-aminoanthracene-induced mutagenic activity under metabolic activation. Additionally, FDD conferred a cytoprotective effect against menadione-induced oxidative stress independent of GSH modulation. The antimutagenic and cytoprotective, as well as its antioxidant benefits, provide additional value to the claimed therapeutic properties of this traditional herbal plant.

## Figures and Tables

**Figure 1 molecules-26-03287-f001:**
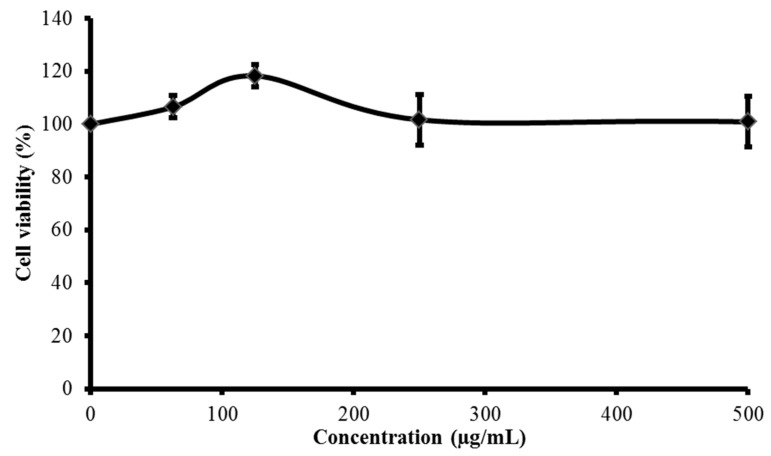
Cytotoxic effect of FDD on V79 cells. V79 cells were treated with various concentrations of FDD (62.5–500 µg/mL FDD) for 24 h. Cell viability was assessed using MTT assay. Each point shown is mean ± SEM of three independent experiments.

**Figure 2 molecules-26-03287-f002:**
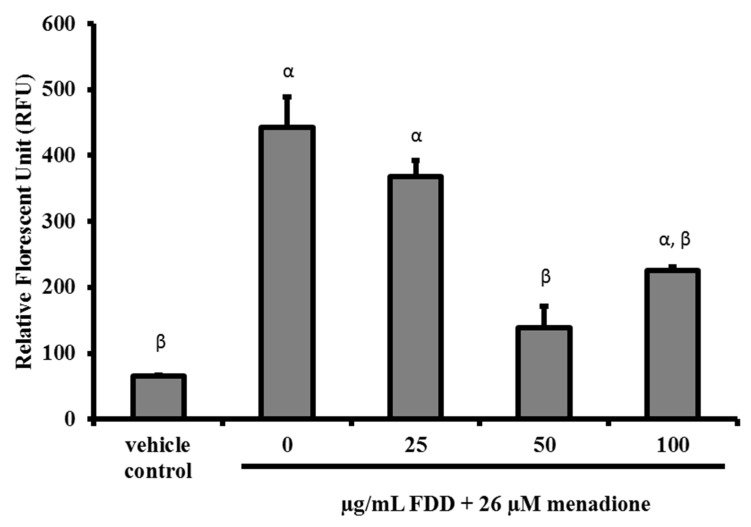
Intracellular superoxide anion level in different treatment groups. V79 cells were pre-treated with various concentrations of FDD for 24 h prior to exposure to 26 µM menadione for 3 h. The treated cells were stained with hydroethidine and analyzed using a flow cytometer. Data were demonstrated as mean ± SEM for three independent experiments. ^α^ Significant difference compared with vehicle control group (*p* < 0.05). ^β^ Significant difference compared with 0 µg/mL FDD + 26 µM menadione group (*p* < 0.05).

**Figure 3 molecules-26-03287-f003:**
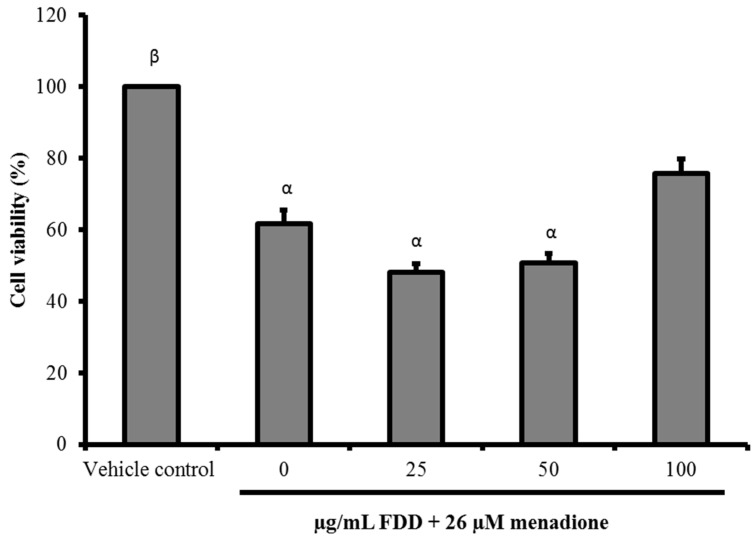
Protective effect of FDD against menadione-induced cell death. V79 cells were pre-treated with various concentrations of FDD for 24 h prior to exposure to 26 µM menadione. Cell viability was determined using MTT assay. Data were demonstrated as mean ± SEM for three independent experiments. ^α^ Significant difference compared with vehicle control group (*p* < 0.05). ^β^ Significant difference compared with 0 µg/mL FDD + 26 µM menadione group (*p* < 0.05).

**Figure 4 molecules-26-03287-f004:**
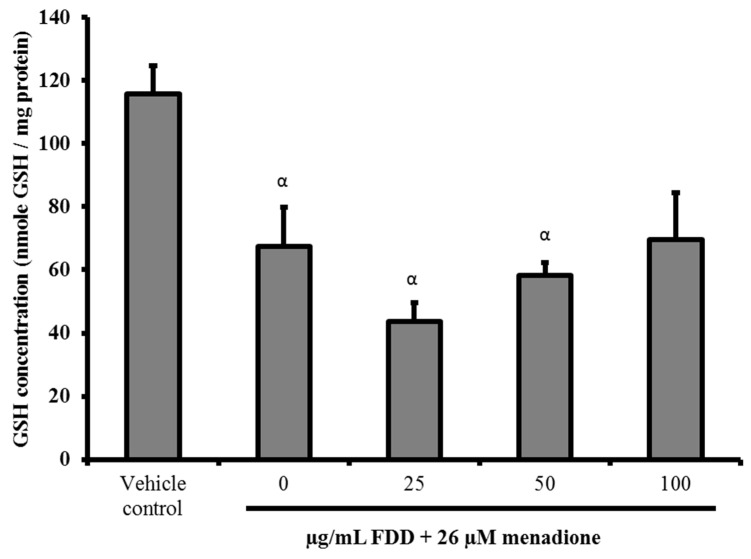
Intracellular GSH level in different treatment groups. V79 cells were pre-treated with various concentrations of FDD for 24 h prior to exposure to 26 µM menadione for 3 h. The treated cells were harvested, and glutathione concentration was measured. Data were demonstrated as mean ± SEM for three independent experiments. ^α^ Significant difference compared with vehicle control group (*p* < 0.05). ^β^ Significant difference compared with 0 µg/mL FDD + 26 µM menadione group (*p* < 0.05).

**Figure 5 molecules-26-03287-f005:**
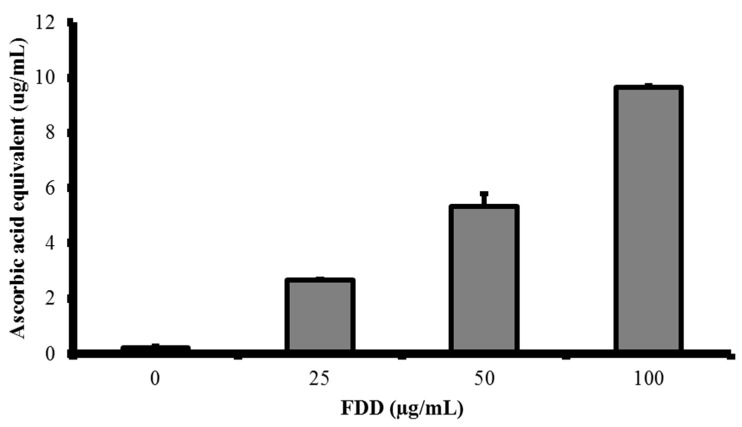
Ascorbic acid equivalent value in different concentrations of FDD. FDD antioxidant capacity was measured using FRAP assay.

**Table 1 molecules-26-03287-t001:** Results of mutagenicity testing of FDD on *Salmonella typhimurium* TA 98 and TA 100 strains with and without metabolic activation.

S. typhimurium Strains	Concentration (mg/mL)	No. of Revertant Colony (Mean ± SD)	
		Without S9	With S9
**TA 98**	PC	444 ± 12.9 ^a, α^	531 ± 19.86 ^b, α^
	NC	14 ± 3.61	19 ± 2.65
	3.125	14 ± 1.00	16 ± 0.58
	6.25	9 ± 0.58	ND
	12.5	10 ± 1.00	20 ± 0.00
	25	14 ± 0.58	ND
	50	17 ± 2.00	19 ± 0.00

**TA 100**	PC	932 ± 32.72 ^c, α^	1209 ± 50.06 ^b, α^
	NC	84 ± 14.18	83 ± 3.21
	3.125	83 ± 4.93	80 ± 6.81
	6.25	63 ± 2.65	ND
	12.5	76 ± 3.06	78 ± 4.00
	25	63 ± 1.73	ND
	50	74 ± 0.58	84 ± 1.00

NC*—*Sterile normal saline; PC*—*Positive control; ND*—*Not defined. Data were shown as mean ± SD of three independent experiments. ^a^ 2-nitrofluorene; ^b^ 2-aminoanthracene; ^c^ sodium azide; ^α^ Significant difference compared with negative control (*p* < 0.05).

**Table 2 molecules-26-03287-t002:** Result of antimutagenicity testing of FDD on *Salmonella typhimurium* TA 98 and TA 100 strains with and without metabolic activation.

Strains	Concentration (mg/mL)	Percentage Inhibition (%)					
		2-Nitrofluorene		2-Aminoanthracene		Sodium Azide	
		−S9	+S9	−S9	+S9	−S9	+S9
TA 98	12.5	11.27 ^α^	−	−	60.25 ^α^	−	−
	50	28.74 ^α^	−	−	74.75 ^α^	−	−
TA 100	12.5	−	−	−	17.91 ^α^	11.27	−
	50	−	−	−	79.73 ^α^	13.9	−

Data were shown as mean ± SD of three independent experiments. ^α^ Significant difference compared with negative control (*p* < 0.05).

## Data Availability

The data presented in this study are available on request from the corresponding author.

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
