# Peer review of "Antimutagenic, Cytoprotective and Antioxidant Properties of Ficus deltoidea Aqueous Extract In Vitro"

_molecules, 2021, doi:10.3390/molecules26113287_

Round 1

Reviewer 1 Report

The subject of the manuscript is very interesting and with possible practical implementation. I would recommend the publication of this work for printing

Author Response

Response to Reviewer 1 Comments

Point 1: The subject of the manuscript is very interesting and with possible practical implementation. I would recommend the publication of this work for printing.

Response 1: Thank you for the comment.

Reviewer 2 Report

This research provides sufficient background and research design, includes relevant references, however some references are quite old. Methods are clear and informative, results and conclusions are clearly presented. English language and style are fine. This research is also valuable in that the study plant grows not only in Asia but also in other parts of the world, which increases the interest of this research for readers. In my opinion, the manuscript is suitable for publishing as it is.

Author Response

Response to Reviewer 2 Comments

Point 1: This research provides sufficient background and research design, includes relevant references, however some references are quite old. Methods are clear and informative, results and conclusions are clearly presented. English language and style are fine. This research is also valuable in that the study plant grows not only in Asia but also in other parts of the world, which increases the interest of this research for readers. In my opinion, the manuscript is suitable for publishing as it is.

Response 1: Thank you for the comment. We have updated some of the references to newer references in the reference list.
